# Serum lipopolysaccharide neutralizing capacity in ischemic stroke

**Jaakko Leskelä**[1], **Milla Pietiäinen**[1], **Anton Safer**[2], **Markku Lehto**[3], **Jari Metso**[4], **Ernst Malle**[5], **Florian Buggle**[6], **Heiko Becher**[7,8], **Jouko Sundvall**[9], **Armin J. Grau**[6], **Pirkko J. Pussinen**[1☯]*, **Frederick Palm**[10☯]

**1** Oral and Maxillofacial Diseases, University of Helsinki, Helsinki, Finland, **2** Institute of Global Health, University of Heidelberg, Heidelberg, Germany, **3** Folkhälsan Institute of Genetics, Folkhälsan Research Center, Helsinki, Finland, **4** Minerva Foundation Institute for Medical Research, Helsinki, Finland, **5** Division of Molecular Biology and Biochemistry, Gottfried Schatz Research Center, Medical University of Graz, Graz, Austria, **6** Department of Neurology, Klinikum Ludwigshafen, Ludwigshafen, Germany, **7** University Medical Center Hamburg-Eppendorf, Institute of Medical Biometry and Epidemiology, Hamburg, Germany, **8** University Hospital Heidelberg, Institute of Global Health, Heidelberg, Germany, **9** National Institute for Health and Welfare, Helsinki, Finland, **10** Department of Neurology, Helios Klinikum Schleswig, Schleswig, Germany

☯ These authors contributed equally to this work.

* pirkko.pussinen@helsinki.fi

**Data Availability Statement:** Due to the limited number of participants and sensitive patient information, the ethics committee of the Landesärztekammer Rheinland-Palatinate has recommended not to publicly deposit the data. The

## Abstract

### Introduction

Periodontitis is associated with increased serum lipopolysaccharide (LPS) activity, which may be one mechanism linking periodontitis with the risk of cardiovascular diseases. As LPS-carrying proteins including lipoproteins modify LPS-activity, we investigated the determinants of serum LPS-neutralizing capacity (LPS-NC) in ischemic stroke. The association of LPS-NC and *Aggregatibacter actinomycetemcomitans*, a major microbial biomarker in periodontitis, was also investigated.

### Materials and methods

The assay to measure LPS-NC was set up by spiking serum samples with *E. coli* LPS. The LPS-NC, LPS-binding protein (LBP), soluble CD14 (sCD14), lipoprotein profiles, apo(lipoprotein) A-I, apoB, and phospholipid transfer protein (PLTP) activity, were determined in 98 ischemic stroke patients and 100 age- and sex-matched controls. Serum and saliva immune response to *A. actinomycetemcomitans*, its concentration in saliva, and serotype-distribution were examined.

### Results

LPS-NC values ranged between 51–83% in the whole population. Although several of the LPS-NC determinants differed significantly between cases and controls (PLTP, sCD14, apoA-I, HDL-cholesterol), the levels did not (p = 0.056). The main determinants of LPS-NC were i) triglycerides (β = -0.68, p<0.001), and ii) HDL cholesterol (0.260, <0.001), LDL cholesterol (-0.265, <0.001), PLTP (-0.196, 0.011), and IgG against *A. actinomycetemcomitans* (0.174, 0.011). Saliva *A. actinomycetemcomitans* concentration was higher [log mean (95%

data requests may be sent to the administration of the ethical committee, email Ethik-kommission@laek-RLP.de.

**Funding:** The study was supported by grants from the Academy of Finland (1266053 for PJP), Finnish Dental Society Apollonia, the Paulo foundation, and the Sigrid Juselius foundation. The funders had no role in study design, data collection and analysis, decision to publish, or preparation of the manuscript.

**Competing interests:** The authors have declared that no competing interests exist.

CI), 4.39 (2.35–8.19) vs. 10.7 (5.45–21) genomes/ml, p = 0.023) and serotype D more frequent (4 vs. 0%, p = 0.043) in cases than controls. Serotypeablity or serotypes did not, however, relate to the LPS-NC.

## Conclusion

Serum LPS-NC comprised low PLTP-activity, triglyceride and LDL cholesterol concentrations, as well as high HDL cholesterol and IgG against *A. actinomycetemcomitans*. The present findings let us to conclude that LPS-NC did not associate with stroke.

## Introduction

The endotoxin, lipopolysaccharide (LPS) is an important virulence factor of gram-negative bacteria. The lipid A integrates LPS into the outer membrane of the bacterial cell and is mainly responsible for the toxicity of the molecule, inducing physiological symptoms of illness, such as fever, muscle aches, and nausea. The O-antigen is the most variable structure of the molecule and a potent antigen in acquired immunity.

The presence of LPS in the circulation, i.e. endotoxemia, is associated with increased risk of cardiometabolic disorders, diabetes, and kidney diseases [1–7]. The recovered LPS-activity in the circulation varies between the moderate increases found in "metabolic endotoxemia" to more than a hundred-fold amplified activities found in sepsis [8, 9]. The main endogenous origin of endotoxins is the gastrointestinal microbiota, while local inflammation or dietary challenge is probably a prerequisite for endotoxemia [5, 10, 11]. For example, periodontitis, which is dysbiosis-driven chronic inflammation in tooth-supporting tissues, may contribute to endotoxemia by bleeding gums and saliva [11, 12]. One of the most studied microbial biomarkers of periodontitis is the serologically heterogeneous *Aggregatibacter actinomycetemcomitans*. The serotypes of this species are characterized by structural differences in the O-antigen of LPS [13]. Diverse serotypes may differ in immunogenicity, which probably plays a role in both periodontitis and cardiovascular diseases [14].

During endotoxemia, LPS can be recovered in bacterial cell walls, fragments of bacterial outer membrane, bound to bacterial or host proteins, or in blood cells [15]. The predominant LPS fraction is associated with circulating plasma lipoproteins [16], especially high density lipoproteins (HDL), which contribute to the detoxification of LPS activity [17]. The distribution of circulating LPS may, however, differ between inflammatory, infectious, or metabolic diseases [18–20]. Therefore, the blood capacity to neutralize LPS activity depends on inflammatory status, lipoprotein and apolipoprotein profile, and concentrations of specific LPS-transferring proteins [21–23], such as phospholipid transfer protein (PLTP), LPS-binding protein (LBP), and soluble (s)CD14 [24]. Previous studies revealed lower LPS-neutralizing capacity (LPS-NC) in patients with alcoholic liver disease [25] or inflammatory bowel disease [26] when compared to controls. In these studies a qualitative gel-clot Limulus Amebocyte Lysate assay (LAL-assay) with increasing amounts of spiked LPS in the sample have been used to measure LPS-NC. While the gel-clot LAL-assay may be considered a useful method, the chromogenic LAL-assay has become the preferential technique to detect LPS-activity.

Endotoxemia is associated with cardiovascular diseases, such as myocardial infarction (MI), coronary artery disease events, and stroke [1–3, 5]. As no data on LPS-NC has been reported in any of these diseases, the present study aimed to investigate LPS-NC in a case-control study of ischemic stroke using the quantitative LAL-assay with a chromogenic substrate.

## Materials and methods

### Population

The present population is a subsample of the larger GenesiS study, a case-control study on genetic, socioeconomic, and infectious determinants of ischemic stroke [27]. Patients between the age of 18 and 80 years with a first-ever ischemic stroke were recruited. The patients with signs of acute infections were excluded. In addition to clinical examination, an interview using a structured questionnaire was conducted and blood samples were collected.

Healthy controls were selected from a random sample of inhabitants living in the coverage area of the registry, based on the official population registry. Persons with previous MI (by self-report) and such with previous stroke according to the population-based stroke register of the Ludwigshafen Stroke Study (LuSSt) were excluded. Controls were examined in parallel to recruitment of patients. Both groups were matched for sex and age (± 2 years). Out of the original 470 stroke patients and 809 matched controls, saliva samples were additionally taken from a subpopulation of consecutive 98 stroke cases and 100 controls. Details including study definitions have been listed previously [28]. Stroke was diagnosed according to the definition of the World Health Organization [29]. The subtype classification was based on brain imaging discriminating between ischemic stroke, intracerebral hemorrhage, or subarachnoid hemorrhage. Stroke etiology was ascertained using modified stroke criteria as previously described [30]. Definition of stroke due to large-artery-atherosclerosis (LAA), cardioembolism (CE), small artery occlusion (SAO), stroke of other determined cause (OTH), and stroke of undetermined cause (UND) was based on TOAST criteria [31]. We additionally addressed "probable atherosclerotic stroke" (AUT) in accordance with the PERFORM study [32]. Serum and paraffin-stimulated saliva samples were collected and information on smoking habit was included from the questionnaire.

The study protocol was approved by the ethics committee of the Landesärztekammer Rheinland-Palatinate and all participants signed an informed consent.

### Serum determinations

Determinations of total cholesterol, triglyceride, HDL cholesterol, apoA-I and apoB concentrations were carried out by using Abbott Architect reagents (Abbott Laboratories, Abbott Park, IL, USA) in an accredited clinical laboratory, Forensic Toxicology Unit at the National Institute for Health and Welfare, Helsinki, Finland. Low density lipoprotein (LDL) cholesterol was calculated by Friedewald formula. For standardizing measurements, the laboratory has taken part in Lipid Standardization Program organized by CDC, Atlanta, USA. During the course of the laboratory measurements of the study samples in 2016, the precision was characterized by the coefficient of variation (mean ± SD) and the accuracy by systematic error (bias) (mean ± SD). Precision and accuracy were 0.6% ± 0.1 and 0.7% ± 0.5 for total cholesterol, 2.0% ± 0.4 and -2.3% ± 2.0 for triglycerides, 1.3% ± 0.3 and 3.0% ± 1.9 for HDL cholesterol, 1.3% ± 0.6 and 2.9% ± 1.5 for apoA-I and 1.7% ± 0.9 and -3.1% ± 1.7 for apoB, respectively.

Phospholipid transfer activity by PLTP was measured using a lipoprotein-independent assay as previously described [33]. LBP and sCD14 concentrations were determined by use of commercial assays from Hycult Biotech (Hycult Biotech Inc, Wayne, PA, USA), catalog numbers HK315-01 and HK320, respectively, according to the manufacturer's suggestions.

### LPS-activity and LPS-neutralizing capacity

LPS-activity was determined using the Limulus Amebocyte lysate assay coupled with a chromogenic substrate (Hycult Ltd) on diluted samples (1:10, v/v of endotoxin-free water)

according to the manufacturer's instructions. The LPS neutralizing assay was established in pilot experiments. An appropriate amount of LPS (*E. coli* 0111:B4, Sigma-Aldrich) to be added in serum was determined in a range of 0.25 ng/ml-16.0 ng/ml and LPS-activity was measured further on. The best sensitivity-specificity was observed, when adding 20 pg of LPS (concentration 5 ng/ml) in serum dilution (5 μl of serum, 4 μl of LPS solution and 41 μl of endotoxin-free water), which was measured to give 3.67 EU/ml (endotoxin units; mean ± SD; 3.6717 ± 0.339 EU/ml; n = 9) in water. This result was used in the calculations of LPS-neutralizing capacity. Then, serum samples of cases and controls were diluted, and divided into two. One aliquot was spiked with the LPS-preparation while the other half aliquot remained untreated. Both were incubated at 37˚C for 30 minutes, and the LPS activities were measured in accordance with the manufacturer's instructions. LPS-NC was calculated as a percentage of neutralized LPS-activity from the theoretical LPS-activity including the original and the added activities:

$$\left(1 - \left[\frac{\text{LPS spiked}}{\text{LPS intact} + 3.67 \frac{EU}{ml}}\right]\right) * 100\%$$

## Saliva determinations

All 198 study participants chewed a piece of paraffin, and at least 2 ml of stimulated whole saliva was collected. Samples were stored at -70˚C until use for the quantitative analyses of both oral bacteria and antibodies binding to them. After thawing, the saliva samples were centrifuged at 9300 *g* for 5 minutes, the supernatants were aliquoted and used for antibody analyses. The pellets were used for quantitative RT-PCR (qPCR) of oral bacteria.

## Quantification of oral bacteria with qPCR and *A. actinomycetemcomitans* serotyping

The saliva pellets were suspended in 200 ml Tris-EDTA buffer and the extraction of total genomic DNA was performed with the ZR Fungal/Bacterial DNA Kit (Zymo Research) according to the manufacturer's instructions. The amounts of *A. actinomycetemcomitans* and *Porphyromonas gingivalis* were determined with qPCR as previously described [28, 34].

The serotype of *A. actinomycetemcomitans* in saliva of the bacterium-positive participants was further examined with *A. actinomycetemcomitans* serotype-specific qPCR assay as previously described [14] with minor modifications. qPCR reactions with total volume of 20 μl contained 1× Kapa SYBR Fast Universal qPCR master mix (Kapa Biosystems, Merck) supplemented with ROX reference dye, 2 μl of salivary DNA and 0.2 μM forward and reverse primers specific to *A. actinomycetemcomitans* serotypes A-E [14]. Each qPCR plate included a 10-fold dilution series of standard DNA extracted from a reference strain representing particular serotype (ATCC 29523 = serotype A, ATCC 43718 = serotype B, ATCC 33384 = serotype C, IDH 781 = serotype D, IDH 1705 = serotype E). qPCR analyses were performed with the Mx3005P Real-Time qPCR System (Stratagene) with following steps: 95˚C for 3 min (initial denaturation), 3 s at 95˚C and 20 s at 60˚C (40 cycles). Dissociation curve was generated according to the default settings of Mx3005P Real-Time qPCR System. The results were analyzed with Strategene MxPro software. Serotypes were identified based on both the amplification plots and the dissociation curves specific to each serotype.

## Measurement of serum and saliva antibody levels against oral bacteria

Serum and saliva antibody levels against *A. actinomycetemcomitans* and *P. gingivalis* were measured from serum and saliva supernatants by multiserotype-ELISA as described previously [35]. The antigens were composed of formalin-killed whole bacteria representing several

serotypes of the species. The strains in the *P. gingivalis* assays were ATCC 33277, W50, and OMGS 434, representing serotypes A, B, and C. The strains in the *A. actinomycetemcomitans* assays were ATCC 29523, ATCC 43718, ATCC 33384, IDH 781, IDH 1705, and C59A representing serotypes A, B, C, D, E, and X (non-serotypeable), respectively. Both IgG- and IgA-class antibody levels were measured. Serum dilutions were 1:100 and 1:200 (*P. gingivalis* IgA/IgG and *A. actinomycetemcomitans* IgA) or 1:1500 and 1:3000 (*A. actinomycetemcomitans* IgA). Saliva dilutions were 1:3.6 and 1:36 for all determinations.

Serum antibody levels against different *A. actinomycetemcomitans* serotypes were determined also separately as described above. Instead of coating the plates with a mixture of reference strains, individual strains were used as antigens. The serum dilutions were 1:100 and 1:200 for IgA determinations, and 1:1500 and 1:3000 for IgG determinations. All levels are presented as ELISA units consisting from mean values of two dilutions determined as duplicates and normalized according to the reference serum applied on each plate. The interassay coefficient of variation was <8% for serotypes A IgA, B IgG, C IgG, <10% for serotypes A IgG, D IgA, D IgG, E IgA, X IgA and <15% for serotypes B IgA, C IgA, E IgG, X IgG.

## Statistical methods

Statistical differences between the groups were tested with two-sided t-test and Mann-Whitney test in continuous and Chi-square test in categorical variables. Variables displaying skewed distribution were logarithmically transformed before analysis. Pearson's correlation coefficient was used for bivariate correlation analyses. Multiple linear regression was used for multivariate model building. Two-sided level of significance was set to 0.05. Data capture was performed using EXCEL, while SAS JMP12 and SPSS v. 24 was used for statistical testing and model building.

## Results

The characteristics of cases and controls as well as the laboratory parameters are presented in Table 1. Serum total cholesterol, HDL cholesterol, apoA-I, and triglyceride concentrations were significantly (p<0.001) lower in cases similarly as PLTP activity (p = 0.001). Surprisingly, also serum LPS-activity (p<0.001) and calculated ratios of LPS with LBP and sCD14 were significantly (p<0.001) lower in stroke cases compared to controls, while LPS-NC did not differ (p = 0.056) between cases and controls. The distribution of serum LPS-NC ranging between 51–83% is presented in Fig 1. sCD14 concentrations were slightly but significantly (p = 0.025) higher in cases than in controls. Dividing the whole stroke group (n = 98) into six subgroups according to stroke etiology revealed no significant differences (Table 1).

Next, we performed correlation analysis to examine relationships between LPS-activity as well as LPS-NC and lipid, apolipoprotein, LPS-transfer proteins, and levels of immunoglobulins to *A. actinomycetemcomitans* in the whole study group (n = 198). Significant positive correlations were found between LPS-activity and triglyceride, total cholesterol, LDL-cholesterol, and apoB concentrations (Table 2). Furthermore, LPS-activity was significantly correlated with PLTP-activity (p = 0.001), but not with other LPS-transferring proteins, LBP or sCD14 (Fig 2). Table 2 further shows significant negative correlations between LPS-NC and triglyceride, total cholesterol, LDL-cholesterol, and apoB concentrations as well LPS-activity (coefficient, p-value: -0.768, <0.001). Positive correlations were found for HDL-cholesterol and IgG to *A. actinomycetemcomitans*. From the LPS-transferring proteins, LPS-NC had a significant, positive correlation only with PLTP activity, but neither with LBP nor sCD14 concentrations (Fig 2).

**Table 1. Characteristics and clinical laboratory values according to stroke status.**

| | Stroke status | | | Stroke etiology[2] | | | | | | |
|---|---|---|---|---|---|---|---|---|---|---|
| | Controls (n = 100) | Cases (n = 98) | P[1] | AUT n = 18 | CEM n = 24 | LAA n = 13 | SAO n = 28 | OTH n = 7 | UND n = 8 | P[3] |
| Age (years) | 69 (67.7–70.4) | 68.2 (66.2–70.1) | 0.350 | | | | | | | |
| Sex (n, % male) | 47 (47.0) | 45 (45.9) | 0.880 | | | | | | | |
| Current smokers (n, %) | 12 (12.0) | 28 (28.6) | **0.004** | | | | | | | |
| LPS (EU/ml) | 1.63 (1.53–1.74) | 1.35 (1.27–1.43) | **<0.001** | 1.22 (1.08–1.39) | 1.35 (1.19–1.52) | 1.57 (1.44–1.72) | 1.39 (1.24–1.57) | 1.17 (0.88–1.55) | 1.32 (1.07–1.62) | 0.194 |
| LPS-neutralizing capacity (%) | 65.8 (64.4–67.1) | 67.8 (66.4–69.2) | 0.056 | 69.1 (65.4–72.8) | 66.2 (63.2–69.2) | 66.2 (62.4–70.0) | 68.6 (65.8–71.3) | 70.3 (64.1–76.5) | 67.5 (60.8–74.1) | 0.507 |
| PLTP (μmol/h/ml) | 5.96 (5.58–6.36) | 5.12 (4.85–5.40) | **0.001** | 5.45 (4.87–6.08) | 4.97 (4.49–5.50) | 5.36 (4.57–6.28) | 4.94 (4.46–5.48) | 4.98 (4.39–5.64) | 5.16 (3.84–6.94) | 0.844 |
| LBP (μg/ml) | 9.04 (8.27–9.87) | 9.79 (9.03–10.62) | 0.184 | 9.71 (8.22–11.5) | 9.2 (7.57–11.2) | 10.4 (8.97–12.0) | 9.42 (8.08–11.0) | 11.5 (10.1–13.1) | 11.0 (7.54–16.2) | 0.736 |
| sCD14 (μg/ml) | 1.61 (1.56–1.67) | 1.72 (1.65–1.79) | **0.025** | 1.64 (1.52–1.77) | 1.63 (1.52–1.76) | 1.91 (1.74–2.11) | 1.70 (1.57–1.85) | 1.89 (1.57–2.27) | 1.76 (1.51–2.05) | 0.160 |
| ApoA-I (g/l) | 1.75 (1.68–1.81) | 1.51 (1.45–1.56) | **<0.001** | 1.53 (1.44–1.64) | 1.48 (1.37–1.59) | 1.49 (1.32–1.68) | 1.50 (1.40–1.60) | 1.65 (1.31–2.09) | 1.48 (1.30–1.67) | 0.801 |
| ApoB (g/l) | 1.05 (1.00–1.11) | 1.02 (0.97–1.06) | 0.284 | 1.00 (0.90–1.11) | 1.05 (0.97–1.14) | 1.04 (0.93–1.18) | 1.01 (0.92–1.10) | 0.91 (0.70–1.18) | 1.05 (0.92–1.20) | 0.747 |
| Cholesterol (mmol/l) | 5.43 (5.20–5.68) | 4.86 (4.68–5.05) | **<0.001** | 4.89 (4.50–5.30) | 4.73 (4.42–5.06) | 4.86 (4.34–5.45) | 4.98 (4.65–5.33) | 4.78 (3.68–6.20) | 5.01 (4.60–5.46) | 0.953 |
| LDL-cholesterol (mmol/l) | 2.14 (1.39–3.29) | 1.32 (0.89–1.96) | 0.109 | 1.45 (0.65–3.26) | 0.87 (0.43–1.74) | 1.19 (0.29–4.85) | 1.91 (0.91–4.01) | 0.80 (0.08–7.89) | 2.09 (0.81–5.44) | 0.700 |
| Triglycerides (mmol/l) | 1.74 (1.58–1.91) | 1.39 (1.28–1.51) | **<0.001** | 1.25 (1.09–1.44) | 1.58 (1.32–1.90) | 1.63 (1.34–1.98) | 1.32 (1.12–1.55) | 1.45 (1.02–2.08) | 1.11 (0.82–1.50) | 0.142 |
| HDL-cholesterol (mmol/l) | 1.32 (1.25–1.38) | 1.11 (1.05–1.17) | **<0.001** | 1.18 (1.08–1.29) | 1.04 (0.94–1.15) | 1.07 (0.93–1.24) | 1.09 (1.00–1.19) | 1.27 (0.90–1.79) | 1.11 (0.88–1.41) | 0.459 |
| LPS / HDL | 1.24 (1.14–1.36) | 1.19 (1.09–1.30) | 0.501 | 1.04 (0.88–1.23) | 1.30 (1.09–1.55) | 1.43 (1.20–1.69) | 1.21 (1.04–1.41) | 0.92 (0.54–1.56) | 1.18 (0.87–1.60) | 0.199 |
| LPS / PLTP | 0.27 (0.25–0.30) | 0.26 (0.25–0.28) | 0.443 | 0.23 (0.19–0.26) | 0.27 (0.24–0.31) | 0.29 (0.26–0.33) | 0.28 (0.25–0.32) | 0.23 (0.17–0.33) | 0.25 (0.17–0.39) | 0.274 |
| LPS / LBP | 0.18 (0.16–0.2) | 0.14 (0.12–0.15) | **<0.001** | 0.13 (0.11–0.15) | 0.15 (0.12–0.19) | 0.15 (0.13–0.19) | 0.15 (0.12–0.18) | 0.10 (0.08–0.12) | 0.12 (0.08–0.18) | 0.371 |
| LPS / sCD14 | 1.01 (0.94–1.09) | 0.79 (0.74–0.84) | **<0.001** | 0.75 (0.66–0.85) | 0.82 (0.71–0.96) | 0.82 (0.70–0.96) | 0.82 (0.73–0.92) | 0.62 (0.49–0.78) | 0.75 (0.56–1.00) | 0.382 |

Presented as mean (95% confidence interval) or number (frequency).

[1]p-values calculated by t-test or Chi-square test.

[2]Stroke subtypes are "probable-atherosclerotic" stroke (AUT), cardioembolism (CEM), large-artery-atherosclerosis (LAA), small artery occlusion (SAO), other (OTH) and undefined (UND) stroke etiology.

[3]p-values calculated by ANOVA.

The association of LPS-activity and LPS-NC with other serum parameters was analyzed with three different linear regression models (Table 3). Triglyceride concentrations (model 1) and further, LPS-activity (model 2) were the strongest determinants of the LPS-NC. In model 3, HDL-cholesterol concentration was positively, and PLTP-activity and LDL-cholesterol concentration negatively associated with the LPS-NC. Neither LBP nor sCD14 associated with LPS-NC. Similarly, the strongest determinants of LPS-activity were triglyceride, total cholesterol, LDL-cholesterol, HDL-cholesterol, and apoB concentrations as well as PLTP-activity, but neither LBP nor sCD14 concentrations.

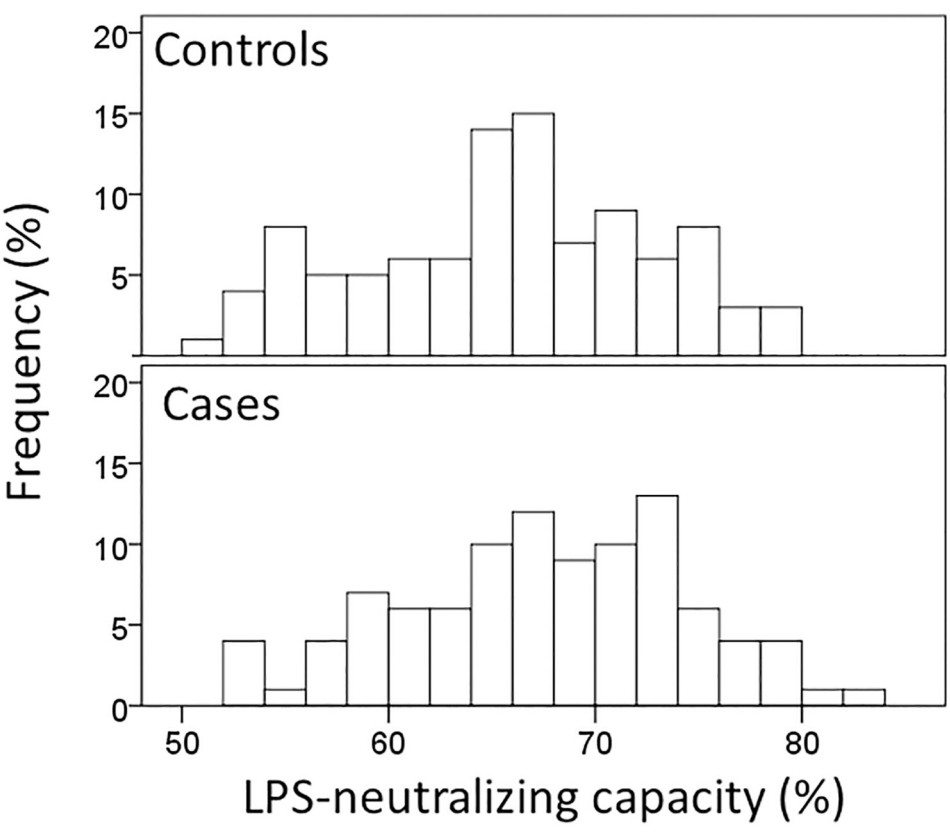

**Fig 1. The distribution of LPS-neutralizing capacity (LPS-NC) of stroke cases and controls.** Serum samples of cases (n = 98) and controls (n = 100) were spiked with 3.67 EU/ml of *E. coli* LPS and the LPS-neutralizing capacity of the serum was calculated as $\left(1 - \left[\frac{\text{LPS spiked}}{\text{LPS intact} + 3.67 \frac{EU}{ml}}\right]\right) * 100\%$.

In addition, the correlations between LPS-activity as well as LPS-NC and two periodontal biomarker species, *P. gingivalis* and *A. actinomycetemcomitans*, were examined. Neither *P. gingivalis* concentration in saliva nor serum or saliva antibody levels against it were significantly correlated with LPS-NC or LPS-activity (p>0.05, S1 Table). However, serum IgG-class

**Table 2. Correlations of LPS-activity and LPS-neutralizing capacity with the clinical laboratory measurements.**

|  | Serum LPS-activity (EU/ml) | Serum LPS-neutralizing capacity (%) |
|---|---|---|
|  | Coefficient (p-value)[1] | |
| Triglycerides (mmol/l) | 0.881 (<0.001) | -0.680 (<0.001) |
| Total cholesterol (mmol/l) | 0.415 (<0.001) | -0.335 (<0.001) |
| HDL cholesterol (mmol/l) | -0.142 (0.050) | 0.153 (0.034) |
| LDL cholesterol (mmol/l) | 0.194 (0.007) | -0.184 (0.010) |
| ApoA-I (g/l) | 0.008 (NS) | 0.009 (NS) |
| ApoB (g/l) | 0.506 (<0.001) | -0.432 (<0.001) |
| *A. actinomycetemcomitans* IgA (ELISA units) | -0.051 (NS) | 0.068 (NS) |
| *A. actinomycetemcomitans* IgG (ELISA units) | -0.113 (NS) | 0.146 (0.040) |

[1]Pearson correlation.

NS, not significant; EU, endotoxin units

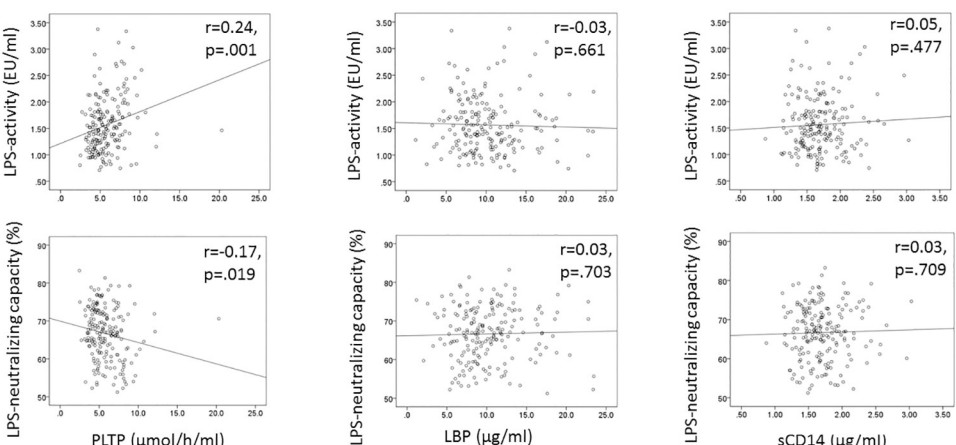

**Fig 2. Scatterplots of LPS-activity and LPS-neutralizing capacity (LPS-NC) with LPS transfer proteins.**
Correlations between serum LPS-NC and LPS-activity with LPS-transfer proteins PLTP, LBP, and sCD14 is shown in the whole study population (n = 198). Correlation coefficients and p-values are presented.

antibody level against *A. actinomycetemcomitans* in a multiserotype-ELISA was a positive predictor of LPS-NC (p = 0.011) (Table 3) and therefore, *A. actinomycetemcomitans* was studied in more detail.

ELISA experiments using individual reference strains representing different serotypes (A to X) separately as antigens revealed that serotypes D (28.0%), E (16.8%), and A (19.7%) were the strongest determinants of the serum IgG-class antibody pool to *A. actinomycetemcomitans* (Fig 3). The percentages were virtually similar for IgA-class antibodies: 24.4%, 16.8%, and 21.0% for serotypes D, E, and A, respectively. The serum IgA-class antibody against serotype C was significantly higher in cases than controls (Table 4), whereas both IgA and IgG solely against the serotype D associated with LPS-NC (Beta 0.385, p = 0.013 and 0.399, p = 0.012). Other serologies neither differed between the cases and controls nor associated with LPS-NC.

**Table 3. Linear regression models for serum LPS activity and LPS-neutralizing capacity.**

| | Dependent variable | |
|---|---|---|
| | **Serum LPS activity (EU/ml)** | **Serum LPS-neutralizing capacity (%)** |
| | **Standardized beta (p-value)** | |
| **Model 1** | | |
| Triglycerides (mmol/l) | 0.881 (<0.001) | -0.680 (<0.001) |
| **Model 2** | | |
| Triglycerides (mmol/l) | 0.736 (<0.001) | 0.063 (NS) |
| LPS (EU/ml) | | -0.844 (<0.001) |
| LPS-neutralizing capacity (%) | -0.250 (<0.001) | |
| **Model 3** | | |
| HDL cholesterol (mmol/l) | -0.251 (0.001) | 0.260 (<0.001) |
| LDL cholesterol (mmol/l) | 0.268 (<0.001) | -0.265 (<0.001) |
| PLTP (μmol/h/ml) | 0.278 (<0.001) | -0.196 (0.005) |
| LBP (μg/ml) | -0.087 (NS) | 0.045 (NS) |
| sCD14 (μg/ml) | 0.022 (NS) | 0.046 (NS) |
| IgG to *A. actinomycetemcomitans* (ELISA units) | -0.146 (0.030) | 0.171 (0.011) |

NS, not significant; EU, endotoxin units

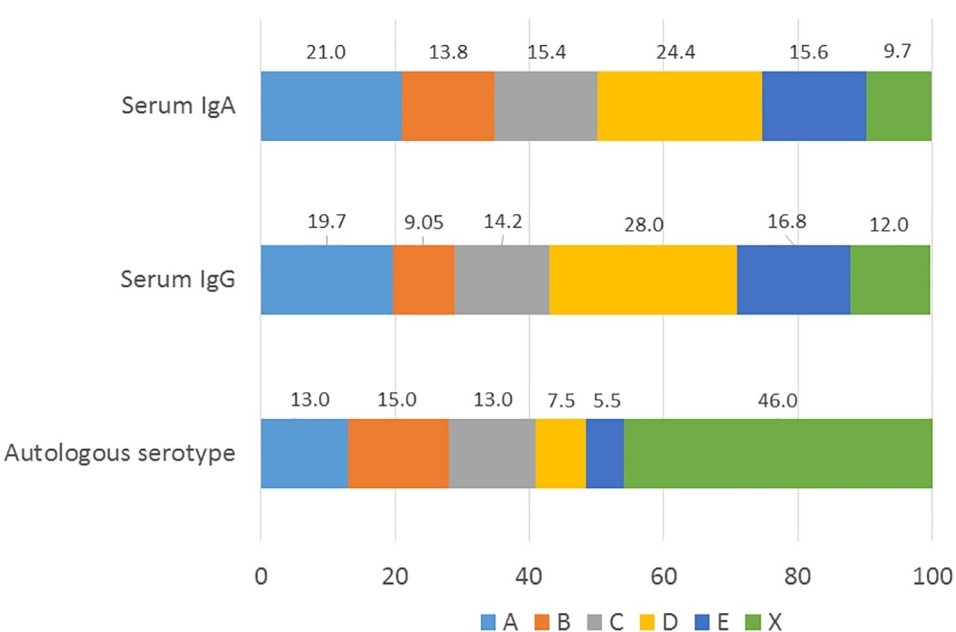

**Fig 3. Relative *A. actinomycetemcomitans* serotype distributions in the population.** Serum IgA- and IgG-class antibodies against different serotypes (A–X) of *A. actinomycetemcomitans* were determined using serotype-specific ELISA. The individual antibody responses to the serotypes were summed up, normalized to 100%, and the proportions of the responses to each serotype were calculated. The autologous serotype was determined from saliva samples of *A. actinomycetemcomitans*–positive subjects (n = 54, 27.3% from the whole population) with qPCR designed for serotyping. The proportions of each detected serotype are shown.

Using qPCR, *A. actinomycetemcomitans* was more frequently detected in the saliva of cases than in controls (36 vs. 19%, p = 0.007) and its mean concentration was higher (p = 0.023) (Table 4). The autologous serotype was determined in the saliva samples using serotype-specific qPCR. Among the 54 (27.3%) *A. actinomycetemcomitans* -positive subjects, serotype was detected in 29 (53.7%) saliva samples. Generally one serotype per subject was detected. The detection rates of the serotypes A to E were 13.0, 15.0, 13.0, 7.5, and 5.5%, respectively (Fig 3). Serotype D was more frequent in cases compared to controls (4 vs. 0%, p = 0.043), otherwise there were no differences.

The serotype-specific antibody levels in serum were examined according to the autologous serotypes in saliva (Fig 4). In serotypes A, B, and C, the serotype-specific serum antibody levels were significantly (p<0.001) higher against the reference strains representing the corresponding serotype. For example, serum IgA-class antibody levels against *A. actinomycetemcomitans* serotype A were highest in those harboring a serotype A strain in saliva compared to those with another *A. actinomycetemcomitans* serotype or no *A. actinomycetemcomitans*. Serotype-ability or serotypes did not, however, relate to LPS-NC (p>0.05, S1 Fig).

## Discussion

We measured a parameter potentially participating in both adaptive and innate immunity responses: serum LPS-neutralizing capacity. Although LPS-NC did not differ between the stroke cases and controls, we were able to determine several parameters predicting LPS-NC levels. In addition to serum lipoproteins, the neutralizing capacity was dependent on PLTP-activity and antibody levels to *A. actinomycetemcomitans*. Notably, neither LBP nor sCD14 concentration predicted LPS-NC or LPS-activity.

**Table 4. _A. actinomycetemcomitans_ and _P. gingivalis_ bacterial levels and antibodies against them.**

| Sample | Species | | | | Stroke | | |
|---|---|---|---|---|---|---|---|
| | | | | | Controls | Cases | p[1] |
| **Serum** | _P. gingivalis_ | Multiserotype | IgA | | 0.53 (0.44–0.64) | 0.54 (0.45–0.64) | 0.894 |
| | | | IgG | | 21.09 (14.8–30.0) | 16.14 (12.6–20.7) | 0.227 |
| | _A. actinomycetemcomitans_ | Multiserotype | IgA | | 1.02 (0.89–1.17) | 1.05 (0.89–1.24) | 0.780 |
| | | | IgG | | 1.12 (1.00–1.26) | 1.07 (0.95–1.20) | 0.582 |
| | | Serotype A | IgA | | 0.73 (0.63–0.85) | 0.86 (0.76–0.98) | 0.110 |
| | | | IgG | | 0.48 (0.41–0.56) | 0.53 (0.48–0.58) | 0.338 |
| | | Serotype B | IgA | | 0.48 (0.41–0.55) | 0.55 (0.48–0.65) | 0.169 |
| | | | IgG | | 0.22 (0.19–0.25) | 0.22 (0.19–0.25) | 0.976 |
| | | Serotype C | IgA | | 0.51 (0.45–0.58) | 0.65 (0.57–0.75) | **0.008** |
| | | | IgG | | 0.34 (0.30–0.39) | 0.36 (0.32–0.40) | 0.521 |
| | | Serotype D | IgA | | 0.91 (0.81–1.03) | 0.92 (0.79–1.07) | 0.923 |
| | | | IgG | | 0.77 (0.68–0.87) | 0.70 (0.63–0.79) | 0.300 |
| | | Serotype E | IgA | | 0.60 (0.52–0.68) | 0.58 (0.50–0.68) | 0.836 |
| | | | IgG | | 0.45 (0.39–0.51) | 0.42 (0.38–0.47) | 0.515 |
| | | Serotype X | IgA | | 0.36 (0.30–0.43) | 0.34 (0.27–0.41) | 0.596 |
| | | | IgG | | 0.33 (0.29–0.38) | 0.29 (0.25–0.32) | 0.114 |
| **Saliva** | _P. gingivalis_ | Multiserotype | IgA | | 0.65 (0.58–0.74) | 0.53 (0.47–0.61) | **0.030** |
| | | | IgG | | 0.09 (0.07–0.13) | 0.05 (0.03–0.07) | **0.018** |
| | _A. actinomycetemcomitans_ | Multiserotype | IgA | | 0.88 (0.78–0.99) | 0.93 (0.82–1.06) | 0.545 |
| | | | IgG | | 0.15 (0.12–0.20) | 0.12 (0.10–0.16) | 0.309 |
| | _P. gingivalis_ | qPCR (genomes/ml) | | | 1.22 (0.41–3.64) | 2.55 (0.86–7.56) | 0.326 |
| | _A. actinomycetemcomitans_ | | | | 4.39 (2.35–8.19) | 10.72 (5.45–21.1) | **0.023** |

Presented as mean (95% confidence interval) of log-transformed values. All antibody levels are expressed as ELISA units.

[1] P-values from the t-test, except Mann-Whitney test for saliva qPCR result.

Spiking experiments of serum with LPS have been commonly used to establish suitable sample processing protocols for the Limulus assay to detect "all LPS-activity present" [23]. _In vivo_, LPS is rapidly neutralized in the circulation [36]. _In vitro_, this neutralizing capacity is present in serum if not inactivated. To investigate the role of different lipoprotein components and LPS-transferring proteins in the neutralization process, we set up in pilot experiments a suitable assay to measure the LPS-NC.

Several reports proposed that LPS not only has a role in atherogenesis but it also may act as a molecular link between microbiome, low-grade inflammation, and cardiovascular diseases [37, 38]. Yet, less attention has been payed to investigate how LPS-NC acts in health and disease. Therefore, we assumed that patients with stroke may have reduced LPS-NC that would eventually lead to LPS-triggered inflammatory cascade, complement activation, contact pathway activation, prooxidative stress, and tissue destruction. However, a difference between stroke cases and controls was not found in the present study. The LPS-preparation used to spike the sample may cause inter-individual differences in the responses [39]; however, due to the relative large population, it was not possible to measure samples spiked with different LPS preparations.

As expected, LPS-NC had a significant positive correlation with HDL-cholesterol, and negative correlations with triglyceride, cholesterol, and LDL-cholesterol concentrations reflecting that LPS supplements the lipid moiety of all lipoprotein subclasses, but that HDL is the most important neutralizer of the LPS-activity [16, 40]. Generally, binding to lipoproteins is the

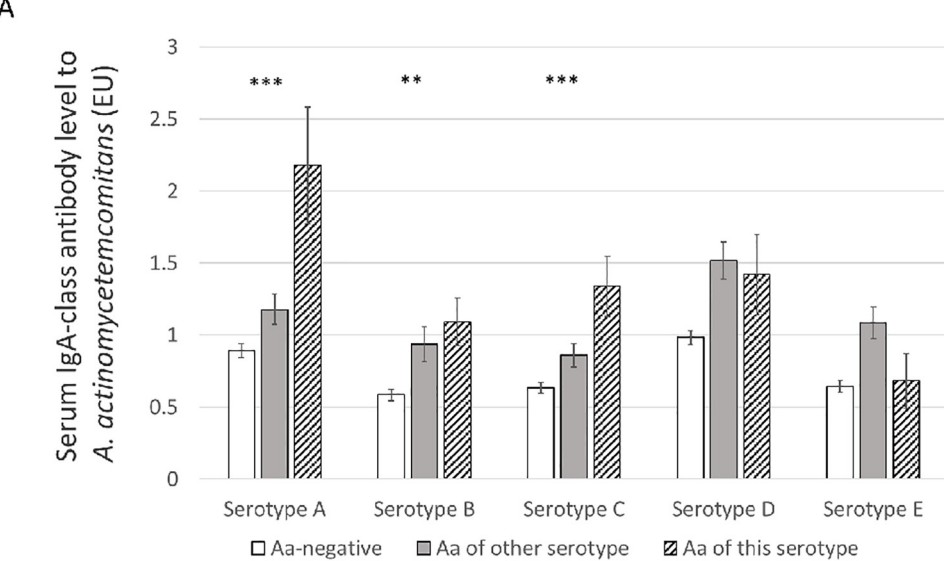

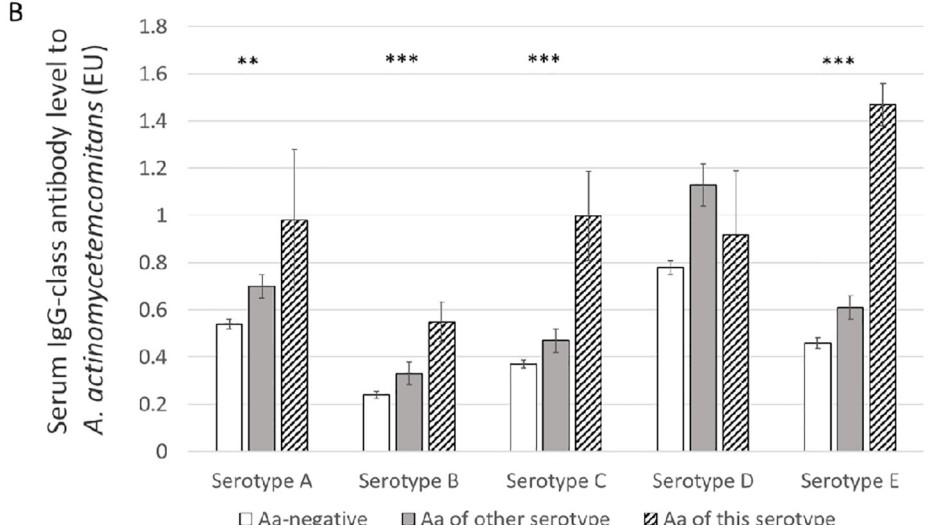

**Fig 4. Serum antibody levels against *A. actinomycetemcomitans* according to the detection of the species and its serotype in saliva.** Serum IgA- (A) and IgG-class (B) antibodies against different serotypes (A–E) of *A. actinomycetemcomitans* were determined by ELISA. Presence of *A. actinomycetemcomitans* in saliva was detected using qPCR, and its serotype was determined by PCR designed for serotyping. The serotype of the antigen in the ELISA is presented on the x-axis separately for those whose saliva did not contain *A. actinomycetemcomitans*, for those having *A. actinomycetemcomitans*, but harbouring different serotype in saliva, and for those harbouring the same serotype in saliva. Number of samples was 198, the columns present the mean values with error bars showing the SE. The asterisks display the terms of linear trend between the groups according to ANOVA-test for logarithmically transformed ELISA-units; *** p<0.001, ** p<0.01.

most important route for LPS detoxification, which thus depends on the lipoprotein subclass distribution [22, 41]. Most importantly, the connection of LPS-NC and lipoprotein subclasses might be bidirectional: the inflammatory response driven by active LPS is accompanied by reduction of lipase activity and induction of triglyceride synthesis leading to increased concentrations of very low density lipoprotein (VLDL) cholesterol and triglycerides [40]. Simultaneously, HDL undergoes conversion resulting in lower HDL-cholesterol concentrations and

decreased removal of peripheral cholesterol to the liver as further consequence [42]. During acute-phase response, these alterations will protect the host from further injury and minimize tissue damage, but in chronic disorders, they may be considered pro-atherogenic. It reasonable to assume that low LPS-NC exposes the subject to persistent inflammatory responses compared to a physiological situation where LPS activity is promptly neutralized. In the present study, LPS-NC had a negative correlation with apoB-concentrations paralleling LDL-cholesterol, but surprisingly no correlation with apoA-I-concentration was observed, although the C-terminal portion of apoA-I is responsible for LPS-neutralization and displays anti-bacterial activity [43, 44].

Several proteins in circulation participate in transferring of LPS: LBP, PLTP, and sCD14 [24]. Compared to healthy controls, higher LBP and sCD14 concentrations have been observed in patients with ischaemic stroke [45], especially in those with a poor short-term prognosis [3]. Similarly, in the present study we observed elevated sCD14 concentration in the stroke cases. A recent study also reported higher levels of CD14+ microparticles to be associated with stroke severity indicating that CD14 may be part of the post-stroke inflammatory process [46]. This might explain our findings with elevated sCD14 concentrations without differences in LPS-NC in stroke cases compared to controls. A long-term follow-up study of the present patient population will reveal, whether any of the parameters are associated with the outcome.

LBP and PLTP have the ability to extract LPS from bacterial outer membrane fragments to HDL particles [24], a process contributing to the clearance of circulating LPS [16, 40]. However, LBP also promotes LPS binding to membrane-bound or soluble CD14, which primarily mediates inflammatory responses [47]. It has been speculated that LPS transfer activity of sCD14 may become more important when serum concentrations of LBP increase and PLTP decrease during acute infection and inflammation [48]. Thus, the relative concentrations of LPS-transferring proteins may have a role in determining the fate of LPS, which plausibly affects the inflammatory response in endotoxemia. However, in the present study neither sCD14 nor LBP concentrations were associated with LPS-NC or LPS-activity.

Among the potent LPS-transferring proteins LPS-NC was associated only with PLTP-activity: stronger neutralizing capacity was found in participants with low PLTP-activity. On the contrary, increased PLTP lipid transfer activity has earlier been measured in both acute and chronic inflammations [49–51]. It has also been shown that PLTP can protect mice from lethal endotoxemia [52] independently of the HDL pool [53] by neutralizing LPS and preventing the growth of gram-negative bacteria [54]. Jänis and co-workers been proposed that PLTP has two forms regarding the lipid transfer activity, an active and an inactive [55]. In the present study, we measured only the active form of PLTP, which was related rather to LPS-activity than LPS-NC. Thus, the multifaceted role of PLTP in infectious and inflammatory disorders deserves further studies.

LPS is a potent antigen and antibodies are generated against all of its main structural components: lipid A, core, and O-antigen. The antibodies are supposed to neutralize LPS-activity, since immune complexes with LPS are rapidly cleared from the circulation [15]. Therefore, it is not surprising that the LPS-NC was positively associated with the antibody levels to *A. actinomycetemcomitans* indicating that these antibodies might neutralize LPS-activity. Since LPS of *A. actinomycetemcomitans* contains serotype-specific antigens [13, 56], we further examined the serology in this population. Especially antibody levels against the serotype D of the species were associated significantly and directly with serum LPS-NC; this serotype was also more frequently found among stroke cases by direct serotyping. Unfortunately, only a few participants harbored serotype D and we were not able to do further comparisons. Serotype D has a distinct O-antigen with repeating tetrasaccharide units, whereas the core and lipid A structure are

more conserved between serotypes [57, 58]. The lipid moiety of *A. actinomycetemcomitans* lipid A has six acyl chains with 14 carbon atoms [57], which is considered having the highest stimulatory effect on human monocytes, but probably not that active in the Limulus assay [59]. Considering all the structural and compositional characteristics affecting the reactivity of any LPS in the Limulus assay [13], the immunogenicity of LPS is impossible to relate to the LPS-activity or its neutralization. Although the serotyping did not further reveal why *A. actinomycetemcomitans* IgG was correlated with LPS-NC, it cannot be considered only as a feature of IgG, since a similar correlation was not observed with IgG binding to *P. gingivalis*.

We previously found that IgA-seropositivity to *A. actinomycetemcomitans* was associated with first-ever ischemic stroke in a larger cohort of the Genesis study [60]. These antibodies have also been found to be associated with increased risk for incident stroke in a 13-year follow-up study [61]. Furthermore, *A. actinomycetemcomitans* leukotoxin-neutralizing antibodies were associated with lower stroke risk in a 5-year follow-up study [62]. However, in the present study, the levels of these antibodies did not differ between cases and controls although the concentration of *A. actinomycetemcomitans* was higher in the saliva samples of the cases. This may be due to the limited sample size (n = 198) which was based on the availability of saliva samples. However, higher concentrations of this *A. actinomycetemcomitans* have been observed also earlier in the oral samples of patients suffering from coronary artery disease [63–65]. *A. actinomycetemcomitans* is not only associated with increased risk for cardiovascular diseases, but may also act as an etiologic microbe in endocarditis [66].

To sum up, we did not find significant differences in the LPS-NC between stroke cases and controls. However, we found that serum LPS-NC is composed of low PLTP-activity, triglycerides and LDL cholesterol, and high HDL cholesterol and IgG binding to *A. actinomycetemcomitans*, whereas apoA-I, LBP, sCD14 or *A. actinomycetemcomitans* serotypes did not associate with LPS-NC.

## Supporting information

**S1 Fig. LPS-neutralizing capacity according to *A. actinomycetemcomitans* serotypeability and serotypes.** LPS-neutralizing capacity (LPS-NC) and *A. actinomycetemcomitans* determinations were performed among 198 subjects. The serotype (A to E) was determined by using qPCR on saliva samples. A) LPS-NC in subjects without the bacterium in saliva, with non-serotypeable bacterium strain, and with a serotypeable strain. B) LPS-NC according to different A. actinomycetemcomitans serotypes. Mean values are shown and the error bars present the standard deviation. P-value is calculated by using the ANOVA-test.
(TIF)

**S1 Table. Correlations of serum LPS and LPS-NC with saliva *P. gingivalis* concentration, and serum and saliva antibody levels against *P. gingivalis*.**
(DOCX)

## Author Contributions

**Conceptualization:** Jaakko Leskelä, Markku Lehto, Ernst Malle, Armin J. Grau, Frederick Palm.

**Formal analysis:** Jaakko Leskelä, Anton Safer, Pirkko J. Pussinen.

**Funding acquisition:** Pirkko J. Pussinen.

**Investigation:** Jaakko Leskelä, Florian Buggle, Heiko Becher, Jouko Sundvall, Frederick Palm.

**Methodology:** Milla Pietiäinen, Markku Lehto, Jari Metso, Jouko Sundvall, Pirkko J. Pussinen, Frederick Palm.

**Resources:** Armin J. Grau, Pirkko J. Pussinen.

**Supervision:** Pirkko J. Pussinen.

**Writing – review & editing:** Milla Pietiäinen, Ernst Malle, Florian Buggle, Armin J. Grau, Pirkko J. Pussinen, Frederick Palm.

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
