## [Decision Letter · Decision Letter 0]

24 Jan 2020

Serum lipopolysaccharide neutralizing capacity in ischemic stroke

PONE-D-19-30948

Dear Dr. Pussinen,

We are pleased to inform you that your manuscript has been judged scientifically suitable for publication and will be formally accepted for publication once it complies with all outstanding technical requirements.

With kind regards,

Alberto G Passi, MD PhD

Academic Editor

PLOS ONE

1. We note that you have included the phrase “data not shown” in your manuscript. Unfortunately, this does not meet our data sharing requirements. PLOS does not permit references to inaccessible data. We require that authors provide all relevant data within the paper, Supporting Information files, or in an acceptable, public repository. Please add a citation to support this phrase or upload the data that corresponds with these findings to a stable repository (such as Figshare or Dryad) and provide and URLs, DOIs, or accession numbers that may be used to access these data. Or, if the data are not a core part of the research being presented in your study, we ask that you remove the phrase that refers to these data.

Additional Editor Comments (optional):

The study is intriguing, well written and very promising. It is an important aspect for the clinical implications.

Reviewers' comments:

Reviewer's Responses to Questions

**Comments to the Author**

1. Is the manuscript technically sound, and do the data support the conclusions?

Reviewer #1: Yes

Reviewer #2: Yes

2. Has the statistical analysis been performed appropriately and rigorously? 

Reviewer #1: Yes

Reviewer #2: Yes

3. Have the authors made all data underlying the findings in their manuscript fully available?

Reviewer #1: Yes

Reviewer #2: Yes

4. Is the manuscript presented in an intelligible fashion and written in standard English?

Reviewer #1: Yes

Reviewer #2: Yes

5. Review Comments to the Author

Reviewer #1: This study inserts into a relevant, complex and constantly evolving topic of study, but it is focused on a very specific and poorly known aspect. The topic is covered in an accurate way. Work premises are highlighted, as well as the current state of knowledge and missing points; the experimental approach is well-conducted and results precisely analyzed, both from a statistical and interpretive point of view, leading to the highlighted conclusions. A well done research involving microbiological and clinical interesting implications.

Reviewer #2: The manuscript by Leskela and coworkers investigates the relationonship between LPS-activty and LPS-neutralizing capacity in ischemic stroke. Although the findings revealed no correlation, authors set up an interesting method to predict LPS-neutralizing capacity.

The manuscript is interesting and all procedures clearly described. The only weakness I found is that the manuscript would be not easily readable by not experts in the field.

6. PLOS authors have the option to publish the peer review history of their article (what does this mean?). If published, this will include your full peer review and any attached files.

Reviewer #1: No

Reviewer #2: No

---

## [Editor Report · Acceptance letter]

12 Feb 2020

PONE-D-19-30948 

Serum lipopolysaccharide neutralizing capacity in ischemic stroke 

Dear Dr. Pussinen:

I am pleased to inform you that your manuscript has been deemed suitable for publication in PLOS ONE. Congratulations! Your manuscript is now with our production department. 

With kind regards,

on behalf of

Prof. Alberto G Passi 

Academic Editor

PLOS ONE